# DRAFT: Dense Retrieval Augmented Few-shot Topic Classifier Framework

**Keonwoo Kim**[*]
Seoul National University
VRCREW Inc.
keonwookim@bdai.snu.ac.kr

**Younggun Lee**
Neosapience
yg@neosapience.com

## Abstract

With the growing volume of diverse information, the demand for classifying arbitrary topics has become increasingly critical. To address this challenge, we introduce **DRAFT**, a simple framework designed to train a classifier for few-shot topic classification. DRAFT uses a few examples of a specific topic as queries to construct Customized dataset with a dense retriever model. Multi-query retrieval (MQR) algorithm, which effectively handles multiple queries related to a specific topic, is applied to construct the Customized dataset. Subsequently, we fine-tune a classifier using the Customized dataset to identify the topic. To demonstrate the efficacy of our proposed approach, we conduct evaluations on both widely used classification benchmark datasets and manually constructed datasets with 291 diverse topics, which simulate diverse contents encountered in real-world applications. DRAFT shows competitive or superior performance compared to baselines that use in-context learning, such as GPT-3 175B and InstructGPT 175B, on few-shot topic classification tasks despite having 177 times fewer parameters, demonstrating its effectiveness.

## 1 Introduction

With the prevalence of the Internet and social media, there is a significant demand for classifying or detecting texts related to specific topics within the vast amount of information pouring in from the Internet. For instance, on social media platforms where an overwhelming volume of content is generated, there may exist a need to monitor and filter content associated with particular issues (e.g., drugs). Additionally, with the remarkable progress in large language models (LLMs) (Brown et al., 2020; Ouyang et al., 2022; Chowdhery et al., 2022; Rae et al., 2021; Scao et al., 2022; Thoppilan et al., 2022) in recent times, there is also an increasing

---
[*]Work done while at Neosapience.

demand to detect specific topics within the text generated by LLMs. The use of LLMs often contains ethical concerns with the issues of hallucination, as they possess the capability to generate morally inappropriate or unintended content (Ganguli et al., 2022; Perez et al., 2022). However, as the amount of content shared on social media and generated by LLMs increase exponentially, verifying each piece of content becomes challenging for individuals.

In natural language processing (NLP), research related to the challenges mentioned earlier can be considered a topic classification task since the demand for automatically classifying texts on a specific topic exists. Recent pre-trained language models (Devlin et al., 2018; Liu et al., 2019; Clark et al., 2020; He et al., 2020) have gained considerable recognition for their ability to achieve high performance in topic classification tasks.

To train a topic classification model, it is common practice to rely on supervised learning using a training dataset with labeled data. Most existing methods on a topic classification have primarily relied on benchmark datasets (Zhang et al., 2015; Auer et al., 2007). To the best of our knowledge, they have predominantly focused on improving model performance on benchmark datasets that involve a limited number of topics rather than addressing long-tailed arbitrary topics in real-world scenarios. Regarding real-world applications, the availability of labeled datasets that cover diverse topics is often limited due to the substantial cost of building such datasets. This constraint poses a challenge for directly deploying topic classification models in practical scenarios, thereby calling for a solution that enables their flexible application.

Few-shot classification, which performs classification using few text examples, can be applied even without a training dataset for a predefined topic, enabling its extension to tasks classifying a diverse range of arbitrary topics. To the best of our knowledge, existing few-shot classification

methods operate exclusively on tasks with two or more defined classes and cannot conduct in one-class classification tasks (Zhang et al., 2023; Chen et al., 2022a; Mukherjee and Awadallah, 2020; Sun et al., 2019). However, by leveraging LLMs with in-context learning (Brown et al., 2020; Holtzman et al., 2021; Min et al., 2021), we can conduct one-class few-shot topic classification tasks across various topics, showcasing superior performance. This approach garnered significant attention due to its applicability in scenarios with limited labeled data (Zhao et al., 2021; Holtzman et al., 2021; Min et al., 2021). However, successful implementation of in-context learning relies on LLMs with billions of parameters (Kaplan et al., 2020a). Such models suffer from high computational costs, extensive resource requirements, and slow inference speed to apply in real-world applications.

To address the few-shot topic classification more efficiently in real-world applications, we propose a simple framework called Dense Retrieval Augmented Few-shot Topic classifier framework (**DRAFT**), which can classify arbitrary topics given limited labeled data. DRAFT uses a dense retriever model (Karpukhin et al., 2020) to construct Customized dataset, using examples of a target topic provided as queries. Subsequently, a pre-trained language model is finetuned with the Customized dataset for the topic classification task. We evaluate the performance of DRAFT by conducting experiments on general benchmarks and manually constructed datasets with 291 topics. The former datasets are widely regarded as benchmark datasets in recent classification research. In contrast, the latter datasets are manually constructed to replicate real-world scenarios, allowing for a thorough assessment of the diverse topic classification capability. They are used for one-class classification tasks where only a single topic is defined. Our experiment results demonstrate that DRAFT consistently achieves competitive or superior performance compared to LLMs, such as GPT-3 175B and Instruct-GPT 175B, which have 177 times more parameters on topic classification tasks. These findings provide strong empirical support for the efficacy of DRAFT in tackling few-shot topic classification tasks. The contributions are summarized as follows:

1. We propose DRAFT as a simple but effective framework that classifies texts related to arbitrary topics using a few labeled data.

2. To the best of our knowledge, we are the first

to attempt to classify a topic in texts through a dense retriever model.

3. We introduce the MQR algorithm, which is the first to accommodate multiple queries simultaneously as inputs for the retriever.

4. The results of extensive experiments show that DRAFT achieves competitive performance compared to large language models.

## 2 Related Works

**Retrieval-augmented methods** Information retrieval aims to find semantically relevant documents based on a query. Traditional methods employ lexical approaches to retrieve support documents using sparse vectors (Hiemstra, 2000; Robertson et al., 2009). With the recent development of deep learning (Vaswani et al., 2017), neural network-based methods have demonstrated high performance. The bi-encoder structure (Karpukhin et al., 2020; Izacard et al., 2022a) offers the advantage of pre-encoding document candidates in an offline setting, which allows for faster computation. Nonetheless, the lack of token-level interaction between query and document tokens in the bi-encoder models can lead to lower performance compared to the cross-encoder models (Devlin et al., 2018). Nevertheless, we use a bi-encoder in DRAFT for efficient retrieval, which facilitates the immediate creation of a classifier for any given topic.

There exist various retrieval-augmented methodologies (Cai et al., 2022; Izacard et al., 2022b) in recent NLP research. They encompass solutions for tasks such as fact retrieval (Thorne et al., 2018), open-domain question answering (Chen et al., 2017; Izacard and Grave, 2020; Lewis et al., 2020; Guu et al., 2020), and others. They also include techniques applied during inference to reduce perplexity in language modeling (Khandelwal et al., 2019) and strategies operate akin to memory for specific knowledge or dialogues (Chen et al., 2022b; Fan et al., 2021). All existing retrieval-augmented methods universally handle only a single query as the input to the retriever and subsequently execute downstream tasks. However, unlike existing approaches, DRAFT processes multiple queries simultaneously as input for retrievers.

**In-context learning with LLMs** Recent LLMs (Thoppilan et al., 2022; Zhang et al., 2022; Scao et al., 2022; Rae et al., 2021; Shoeybi et al.,

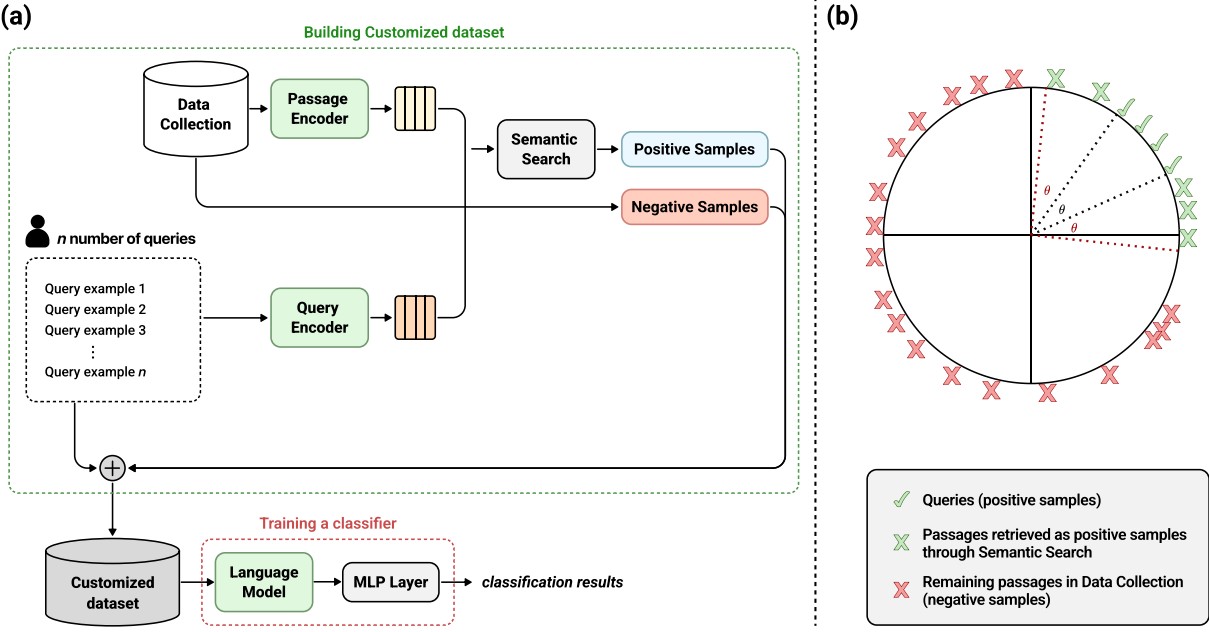

Figure 1: (a) Overall pipeline of DRAFT. DRAFT receives $n$ queries as input, and a trained classifier is only used in the test phase. (b) Illustration of MQR in two-dimensional space. A circle represents the normalized embedding space of texts in Data Collection. For each query, passages only within an angle size $\theta$, calculated as a threshold from $n$ query vectors, are retrieved as positive samples, while others are classified as negative samples.

2019; Chowdhery et al., 2022) represent a critical development in NLP and have been considered an attempt to develop intelligent language systems with fluency approaching that of humans. According to (Kaplan et al., 2020b), as the scale of the language model increases, its performance is also improved on many tasks that typically require models with hundreds of billions of parameters. LLMs primarily perform tasks through in-context learning (ICL) (Brown et al., 2020; Holtzman et al., 2021; Min et al., 2021), which feeds concatenated prompt and $k$ input-target examples (referred to as '$k$-shot') into the model without weight updates. It exhibits superior performance over zero-shot inference across an extensive array of tasks (Zhao et al., 2021; Liu et al., 2021a). For classification tasks, models predict a label from a predefined set of labels with the highest probability. ICL offers the key benefit of allowing a single model to instantly handle multiple tasks with only a limited number of labeled examples, known as few-shot learning. In the context of our research on few-shot topic classification across diverse topics, creating a classifier for each topic without having a training dataset can be regarded as a distinct task. Thus, we use ICL with LLMs, which can perform diverse tasks with only a few labeled examples, as baseline models in the experiments.

## 3 Method

In this section, we begin by describing the application of DRAFT to a simple binary classification and elaborate on the extension of DRAFT. DRAFT comprises two stages for few-shot topic classification: (1) constructing Customized dataset from multiple queries using a dense retriever model and (2) training a classifier. Figure 1 (a) illustrates the overall process of DRAFT.

### 3.1 Building Customized dataset

In the first stage, we use a pre-trained dense retriever, a bi-encoder consisting of a query encoder and a passage encoder, to construct *Customized dataset*. Customized dataset is created by employing a few texts related to a target topic as queries. To efficiently retrieve relevant passages from Data Collection, which serves as the external knowledge base (e.g., Wikipedia), we employ a Maximum Inner Product Search (MIPS) algorithm (Shrivastava and Li, 2014) that finds the vector with the highest inner product value with a given query vector. The retriever is defined by employing a query encoder $E_{query}$ and a passage encoder $E_{passage}$:

$$p(z \mid x) \propto \exp\left(E_{query}(x)^T E_{passage}(z)\right).$$

$E_{query}$ embeds query $x$, and $E_{passage}$ embeds passages $z \in Z$, where $Z$ indicates Data Collection.

We select the top-$k$ passages with the highest prior probability $p(z \mid x)$, which is proportional to the inner product of the query and passage vectors.

---

**Algorithm 1:** Multi-Query Retrieval

**Result:** An array $C$ of target samples
**Input**: An array $S$ with $n$ queries
$\cos(\theta) \leftarrow \frac{1}{nC_2} \sum_{i=1}^{n} \sum_{j \neq i}^{n} sim(S[i], S[j])$
**for** $q = 1, 2, \ldots, n$ **do**
    retrieve $k$ passages using MIPS
    **for** $passage = 1, 2, \ldots, k$ **do**
        $\alpha \leftarrow cos(S[q], passage)$
        $cos$ denotes cosine similarity function
        **if** $cos(\theta) \leq \alpha$ **then**
            $C \leftarrow C + [passage]$
        **end**
    **end**
**end**

---

We define *Customized dataset* as the construction of a collection of positive and negative samples, including multiple queries, specifically designed to train a topic classifier. To build positive samples within Customized dataset, we propose the Multi-Query Retrieval (MQR) algorithm, depicted in Figure 1 (b). As outlined in Algorithm 1, MQR begins by gathering $n$ sentences or keywords related to the specific topic to form queries. A dense retriever model for each query is employed to retrieve top-$k$ passages, where $k$ denotes the subspace size. From the retrieved $n \times k$ passages, only passages vectors that exceed a threshold $\cos(\theta)$ determined by the average pairwise cosine similarity score among the $n$ query vectors are retained. Unlike general dense retriever models that take a single query as input, it can accept multiple queries as input. By combining the $n$ queries and the $m$ passages retained from Data Collection, we collect a total of $n + m$ positive samples. Subsequently, we create an equivalent number of negative samples by randomly selecting passages from Data Collection, thereby forming Customized dataset, which comprises $2(n + m)$ samples.

### 3.2 Training a topic classifier in DRAFT

In the following stage, we proceed with fine-tuning a pre-trained language model on the classification task, employing Customized dataset constructed in the preceding stage. The fine-tuning process is similar to (Devlin et al., 2018) for downstream tasks. It passes [CLS] token, a special classification token representing sentence-level embedding obtained from the hidden vector after passing through the encoder layers through an MLP layer. The classifier is trained using binary cross-entropy loss.

$$L = -\frac{1}{2(n+m)} \sum_{i=1}^{2(n+m)} \sum_{j \in \{0,1\}} \mathbf{1}_{y_i=j} \ln q(y_i \mid x_i),$$

with an indicator function $\mathbf{1}$, a text $x_i$, a label for the corresponding text $y_i \in \{0, 1\}$, and a probability on model's prediction q.

### 3.3 Expanding the Capabilities of DRAFT

**Negative query** We can improve DRAFT's performance by introducing an additional stage after training a classifier. In this stage, we manually incorporate negative queries that belong to a semantically similar category but are different from the target topic. Constructing negative samples follows the process outlined in Section 3.1 for constructing positive samples, which uses MQR. After training the classifier with the updated Customized dataset, it becomes more robust in handling hard negatives that are difficult to classify. This expansion of DRAFT enables it to handle complex cases and enhance its overall performance.

**Multi-class classification** Moreover, through the expansion of the class set, it is possible to develop a multi-class classifier. For each class, MQR is employed to construct class-specific positive samples. The training dataset for a multi-class classifier is created by merging the aggregated class-specific constructed positive samples for each class, eliminating the need for a separate process of constructing negative samples within each class due to the presence of independent other defined classes. Subsequently, the classifier is trained with the merged dataset using cross-entropy loss.

## 4 Experiments

To evaluate the performance of DRAFT, we conduct three experiments with limited labeled data in classification tasks.

1. We assess the performance of DRAFT in few-shot topic classification by manually constructing datasets comprising 291 unique topics that simulate real-world scenarios.

2. We explore the expanding capabilities of DRAFT using negative queries on two additional manually constructed datasets.

3. We evaluate the multi-class classification capabilities of DRAFT using three commonly used benchmark datasets.

## 4.1 DRAFT Setup

We use a Wikipedia dump from 2018 for Data Collection. All encoders, a bi-encoder, and a classifier used in DRAFT are initialized with SimCSE (Gao et al., 2021) trained through supervised learning. While each encoder shares the same weights and contains approximately 330 million parameters, these weights are not shared during the training of DRAFT. We first train the bi-encoder in DRAFT on Natural Question (Kwiatkowski et al., 2019) using contrastive InfoNCE loss (Oord et al., 2018). Once trained, the weights of the bi-encoder remain fixed, only requiring fine-tuning of a classifier when the type of topic is modified. Since the number of samples extracted from MQR varies for each dataset, the size of Customized dataset used by a classifier in DRAFT for training also fluctuates accordingly.

When fine-tuning a classifier, we use the AdamW (Loshchilov and Hutter, 2017) optimizer with a learning rate of 1e-5 using a linear learning rate scheduler. We divide Customized dataset into an 80% to 20% split to form the training and validation datasets, respectively. The classifier is trained by employing early stopping against the validation loss with the patience of two epochs. We perform all experiments using the PyTorch framework and three 32GB Tesla V100 GPUs.

## 4.2 Few-shot topic classification task

**Dataset** To evaluate the effectiveness of DRAFT in few-shot topic classification, specifically in diverse topic scenarios, we construct 291 test datasets. Given the absence of existing benchmark datasets specifically designed for its task, we perform web crawling on the FactsNet website[1], which covers a broad spectrum of subjects comprising 291 distinct topics. The content on the website has a three-level hierarchical structure consisting of a major category, a subcategory, and a subtopic. The major categories are comprised of five types: Lifestyle, History, Nature, World, and Science. Additional information about datasets is in Appendix A.

In our dataset construction, we leverage contents in subtopics. We consider classifying each subtopic as a distinct topic classification task, resulting in 291 test datasets. We select subtopics with at least

---

[1]The website is as follows: https://facts.net

50 samples that belong to the positive class. For each dataset, we construct negative samples, composed of easy negatives and hard negatives, to the same size as the positive samples. Easy negatives are randomly sampled from subtopics that belong to a different subcategory than the positive ones; hard negatives are randomly mined from subtopics within the same subcategory as the positive class.

**Baselines** We set LLMs as baseline models due to their capability to serve as classifiers for various topics when provided with a few labeled samples for ICL. The format of ICL is described in Appendix B.3. We use two types of LLMs, GPT-3 (Brown et al., 2020) and InstructGPT (Ouyang et al., 2022), both with two versions of model sizes: 2.7B and 175B. InstructGPT is an improved version of GPT-3, fine-tuned through reinforcement learning with human feedback to enhance its understanding of prompts.

**Detail settings** Given the absence of training datasets, we adopt Quick Facts, a collection of five texts associated with each subtopic in FactsNet, as queries for DRAFT and examples of ICL, serving as sample instances of the positive class. We employ various ICL examples from Quick Facts for LLMs, specifically selecting one, three, or five examples. For DRAFT, we configure a set of five examples as queries, with a subspace size of 10,000 allocated for each query. The examples of Quick Facts can be seen in Appendix B.4.

To assess the capability of classifying diverse topics, we derive rankings based on the F1 score for each major category across all methodologies and subsequently compute an average rank based on the five major categories. The robustness of each methodology in topic classification can be assessed through the average rank.

**Results and analysis** Table 1 presents the evaluation results of the few-shot classification tasks on diverse topics. We evaluate the F1 score for all 291 subtopics and aggregate the results based on the five major categories. Among the baselines with billions of parameters, except for InstructGPT 175B 1-shot, DRAFT, only with millions of parameters, demonstrates superior performance compared to the others in all categories. When considering the average rankings across five major categories, DRAFT achieves the highest rank of 1.4, followed by InstructGPT 175B 1-shot with an average rank of 1.6, implying DRAFT's optimality.

| Major category | DRAFT | GPT-3 2.7B | | | GPT-3 175B | | | InstructGPT 2.7B | | | InstructGPT 175B | | |
|---|---|---|---|---|---|---|---|---|---|---|---|---|---|
| | | 1-shot | 3-shot | 5-shot | 1-shot | 3-shot | 5-shot | 1-shot | 3-shot | 5-shot | 1-shot | 3-shot | 5-shot |
| Lifestyle | **80.5**(1) | 55.1(4) | 52.7(5) | 48.5(9) | 23.0(13) | 50.3(8) | 56.4(3) | 45.8(11) | 52.2(6) | 47.5(10) | 79.1(2) | 52.6(6) | 35.7(12) |
| History | **84.7**(1) | 53.4(5) | 53.4(5) | 48.8(8) | 23.7(13) | 51.6(7) | 58.2(3) | 40.5(11) | 48.3(9) | 47.8(10) | 82.8(2) | 55.0(4) | 37.1(12) |
| Nature | 79.6(2) | 54.8(7) | 59.7(3) | 56.7(4) | 16.7(13) | 45.7(9) | 55.0(6) | 40.8(11) | 46.9(8) | 39.3(12) | **82.3**(1) | 55.8(5) | 41.0(10) |
| World | 82.1(2) | 54.9(6) | 56.0(5) | 52.6(8) | 17.3(13) | 54.7(7) | 61.2(3) | 41.6(11) | 46.9(9) | 45.4(10) | **86.9**(1) | 59.1(4) | 39.0(12) |
| Science | **80.2**(1) | 54.3(4) | 54.0(5) | 47.2(9) | 20.4(13) | 49.6(7) | 54.6(3) | 44.7(10) | 49.3(8) | 43.3(11) | 79.9(2) | 50.5(6) | 35.2(12) |
| Avg. Rank | **1.4** | 5.2 | 4.6 | 7.6 | 13 | 7.6 | 3.6 | 10.8 | 8 | 10.6 | 1.6 | 5 | 11.6 |

Table 1: F1 scores on few-shot topic classification tasks. We use *k*-shot setting, where *k* denotes the number of examples with labels for ICL on LLMs. We present the average score for each subtopic within its corresponding major category. The values within parentheses indicate the ranking of the 13 methods based on the highest scores for each row. 'Avg. Rank' represents the average ranking for each method.

| | Method | | | |
|---|---|---|---|---|
| Major category | DRAFT | Random | Noun. | Dense. |
| Lifestyle | **80.5**(1) | 50.2(4) | 69.7(2) | 58.0(3) |
| History | **84.7**(1) | 49.4(3) | 69.3(2) | 42.5(4) |
| Nature | **79.6**(1) | 50.8(4) | 62.9(3) | 71.6(2) |
| World | **82.1**(1) | 50.2(3) | 56.7(2) | 45.3(4) |
| Science | **80.2**(1) | 48.5(4) | 54.5(3) | 58.7(2) |
| Avg. Rank | **1.0** | 3.6 | 2.4 | 3 |

Table 2: Results show F1 scores on few-shot topic classification tasks. DRAFT, Noun-based (Noun.), and Dense-based (Dense.) all employ the same 5-shot setting.

**Ablation study**  We conduct an ablation study to highlight the difficulty of few-shot topic classification across diverse topics. We establish three simple baselines in this experiment using the same datasets in Table 1. (1) **Random** method randomly assigns classes to a sample with a uniform distribution of positive and negative classes, where the positive class corresponds to a target topic while the negative class corresponds to all other topics. (2) **Noun-based** method classifies a sample as a target topic if any nouns from the queries exist in the sample. We use NLTK package[2] to extract all nouns. (3) **Dense-based** method classifies a sample using embeddings obtained from [CLS] token. If the cosine similarity score between a query vector and a sample vector surpasses a threshold determined by MQR for any of the queries, the sample is classified as a target topic. The queries used in (2) and (3) are identical to those used in DRAFT.

We observe a clear trend of DRAFT outperforming three baselines across all main categories in Table 2. It implies the difficulty of the task and highlights the limitations of simple approaches, such as Noun-based or Dense-based methods. Nevertheless, the Noun-based method outperforms LLMs

in Table 1, excluding InstructGPT 175B 1-shot. Also, the most straightforward approach, Random method, surpasses some LLMs. Thus, we emphasize that LLMs are ineffective for few-shot topic classification tasks across diverse topics.

### 4.3  Including negative queries on DRAFT

**Dataset**  We manually construct two additional test datasets with Religion and South Korea topics using a similar format to Section 4.2. In contrast to the automatically constructed datasets in Section 4.2 by crawling the website, five annotators manually construct datasets that determine whether each sample belongs to the positive, easy negative, or hard negative class. The easy negative and hard negative in the test dataset share the same negative class label. However, the differentiation is made to demonstrate the construction process of the negative dataset. We define easy negatives as samples unrelated to the positive class in terms of their semantic content. In contrast, hard negatives are samples that fall into semantically similar categories to positive ones but have distinct content. In Religion dataset, we define 'Jewish' and 'Islam' as positive classes. In contrast, the hard negatives consist of content that falls within the religion category but pertains to different specific religions, such as 'Buddha' and 'Hinduism'. The easy negatives are composed of content unrelated to religion altogether. The examples of classes within the dataset on Religion dataset, additional details about South Korea dataset, and instructions for annotators can be found in Appendix A.

**Baselines**  We investigate the impact of different methods on building negative datasets in DRAFT using three distinct methods: M1, M2, and M3. **M1** is defined by its use of random sampling, while **M2** exclusively employs negative queries as dictated by MQR. **M3** is a combination of two methods, with

---

[2]NLTK package is from https://www.nltk.org

50% of the dataset constructed from M1 and the remaining 50% from M2.

**Detail settings**  We use three positive and two negative queries in Religion dataset, whereas four positive and three negative queries in South Korea dataset. The subspace size associated with each query is set at 10,000. Detailed instances of two types of queries can be seen in Appendix B.4.

| Dataset | Method | F1 (%) | Acc (%) | P (%) | EN (%) | HN (%) |
|---------|--------|--------|---------|-------|--------|--------|
| | M1 | 79.0 | 79.5 | **96.8** | **100.0** | 40.0 |
| Religion | M2 | 69.0 | 76.9 | 64.5 | 86.4 | 84.0 |
| | M3 | **95.2** | **96.2** | **96.8** | **100.0** | **92.0** |
| | M1 | 78.1 | 79.5 | 73.2 | **100.0** | 76.5 |
| S.K. | M2 | 72.6 | 69.6 | **80.4** | 18.2 | 85.3 |
| | M3 | **85.4** | **86.6** | 78.6 | **100.0** | **91.2** |

Table 3: Evaluation results on Religion and South Korea (S.K) datasets. Positive (P), easy negative (EN), and hard negative (HN) are class labels classified during the dataset construction process.

**Results and analysis**  We evaluate the F1 score and accuracy for each methodology on the Religion and South Korea datasets. To further examine the impact of negative queries, we measure the accuracy based on the three distinct classes defined during the dataset construction process, which include the positive, easy negative, and hard negative classes. In Table 3, M1 demonstrates proficient classification of easy negatives, albeit struggles in effectively identifying hard negatives. Conversely, M2 exhibits reasonable performance in classifying hard negatives but at the expense of lower accuracy for easy negatives. Remarkably, M3, which builds a negative dataset constructed through a combination of random sampling and employment of negative queries, consistently shows superior performance. Our experimental findings emphasize the significance of M3 in effectively mitigating bias associated with negative queries while ensuring accurate classification of semantically independent content from positive samples. It verifies the potential of employing negative queries to enhance the performance of DRAFT.

### 4.4 Multi-class classification task

**Dataset**  We employ three general benchmark datasets to evaluate the performance of DRAFT with a limited number of samples. AGNews (Zhang et al., 2015) is a collection of news articles used for a 4-way topic classification task. DBpedia (Auer et al., 2007) is an ontology dataset for a 14-way

topic classification task. TREC (Voorhees and Tice, 2000) is a dataset for a 6-way question classification task, which differs from topic-based content. Additional details can be found in Appendix A.

**Baselines**  Among various LLMs, we consider GPT-2 XL with 1.3B parameters (Radford et al., 2019) and GPT-3 (Brown et al., 2020) with different model sizes, such as 175B and 2.7B parameters.

**Detail settings**  We conduct ICL with LLMs to evaluate their performance given $k$ labeled randomly sampled examples from the training dataset for each class, with $k$ being 1, 4, and 8. Similarly, we randomly select eight samples corresponding to each class from the training dataset to serve as queries for DRAFT. Empirically, experiments conducted in Section 4.2 and Section 4.3, which solely rely on one-class samples as few-shot samples, akin to one-class classification, are more challenging compared to multi-class classification, which provides few-shot samples for all classes. Considering the difficulty of the task, we set the subspace size to 50 in this experiment.

| | k-shot | AGNews | DBpedia | TREC |
|---|--------|--------|---------|------|
| | 1 | $45.4_{8.4}$ | $33.6_{18.9}$ | $21.5_{5.2}$ |
| GPT-2 XL (1.5B) | 4 | $44.6_{12.2}$ | $53.0_{14.8}$ | $23.1_{5.9}$ |
| | 8 | $57.1_{11.6}$ | $66.0_{3.6}$ | $32.7_{7.5}$ |
| | 1 | $33.0_{5.1}$ | $25.9_{4.4}$ | $24.3_{6.4}$ |
| GPT-3 (2.7B) | 4 | $43.3_{8.3}$ | $61.0_{12.8}$ | $25.8_{11.5}$ |
| | 8 | $50.8_{7.8}$ | $72.6_{4.5}$ | $29.3_{8.0}$ |
| | 1 | $62.1_{6.3}$ | $79.3_{3.0}$ | $57.7_{6.0}$ |
| GPT-3 (175B) | 4 | $61.0_{10.9}$ | $84.6_{5.8}$ | $\mathbf{60.2}_{7.6}$ |
| | 8 | $79.1_{2.6}$ | $82.3_{7.8}$ | $45.6_{4.0}$ |
| DRAFT | 8 | $\mathbf{82.8}_{1.3}$ | $\mathbf{94.7}_{1.3}$ | $36.0_{6.2}$ |

Table 4: Results show accuracy on benchmark datasets. 'k-shot' represents the number of examples used in ICL. All values are presented in the $mean_{std}$ format.

**Results and analysis**  Experiments are run five times, each iteration using a different random seed for sampling queries and examples. Table 4[3] presents the accuracy results in classification on benchmark datasets. DRAFT outperforms all baselines on the topic classification tasks in both AG-News and DBpedia. Comparing DRAFT with the best performance cases of GPT-3 175B, the results show a difference of 3.7%p accuracy in AGNews and 12.4%p in DBpedia. While DRAFT achieves a lower accuracy than GPT-3 175B in TREC, it still outperforms GPT-2 XL and GPT-3 2.7B.

---

[3]Results on baselines are reported from (Zhao et al., 2021)

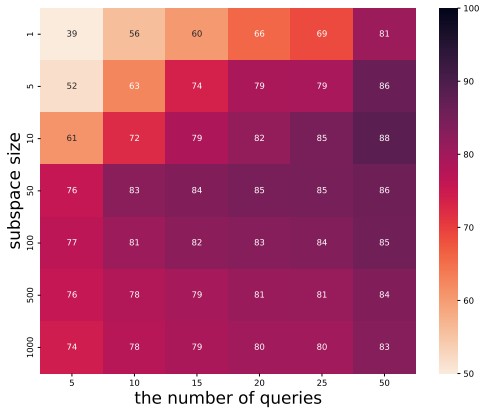

Figure 2: Heatmap shows the impact of the number of queries and subspace size on DRAFT using AGNews.

We find that the performance of DRAFT varies across different benchmark datasets, suggesting that the attribute of Customized dataset plays a crucial role. As DRAFT uses Data Collection to construct Customized dataset, the choice of Data Collection strongly influences its performance. In our experiments, by leveraging Wikipedia as Data Collection, which primarily consists of topic-based content, DRAFT consistently outperforms all baselines with lower variance on AGNews and DBpedia, which also consist of topic-based content. However, on TREC, which involves attributes different from topic-based content, DRAFT exhibits a lower performance compared to GPT-3 175B. These results indicate that DRAFT works outstanding for classifying topics but may not exhibit robust performance in other classification tasks.

## 5 Discussion

**Potential points in DRAFT** In DRAFT, the number of queries and the subspace size are considered the most crucial factors among various hyperparameters. Queries provide information related to the target topic, while subspace size determines how many passages related to each query are retrieved from Data Collection. They influence the performance of DRAFT, as they significantly impact the construction of Customized dataset, which directly impacts the process of training a classifier.

We investigate the relationship between the number of queries and the subspace size by varying both variables using AGNews. The experiments are repeated five times with different configurations, and the average accuracy results are presented in

Figure 2. Increasing the number of queries exhibits a positive correlation with accuracy when the subspace size is fixed. The highest accuracy of 88% is achieved with a subspace size of 10 and 50 queries. Although the increase in the number of queries is limited to 50 due to computational resource constraints, the consistent trend implies that more queries can improve performance.

We find that DRAFT has the potential for improvement with an increased number of queries. Unlike DRAFT, LLMs can suffer from degraded performance due to the majority label bias (Zhao et al., 2021) when the number of examples from the same class increases in few-shot samples. In Section 4.2, InstructGPT 175B demonstrates a noticeable decline in performance as the value of $k$ increases in $k$-shot settings. Considering majority label bias and experimental results, DRAFT shows a more robust performance than LLMs.

**Data Collection** We underscore the importance of Data Collection in Section 4.4. DRAFT shows the highest accuracy on DBpedia due to the similar distribution between the test dataset and Data Collection. However, classifying topics that reflect recent content becomes a challenging issue for DRAFT, which uses a Wikipedia dump from 2018 for Data Collection, since there is a discrepancy between the distribution of the training dataset and that of topics that reflect the recent content. DRAFT can simply solve the problem by building Data Collection with recent knowledge. From the perspective of injecting external knowledge into the model or framework, DRAFT offers a more straightforward approach than LLMs (Meng et al., 2022; Mitchell et al., 2021).

**Real-world applications** DRAFT can effectively classify content related to specific topics from the Internet or SNS. It can swiftly generate a classifier for the given topic upon user request with few queries. However, a challenge emerges as the number of tasks for classifying topics escalates significantly with the increasing number of users on the Internet or the growing quantity of topics requested. Storing the weights of all individually trained classifiers for each task is inefficient. Thus, to effectively implement DRAFT in real-world scenarios, one must consider an approach centered on parameter-efficient learning (Houlsby et al., 2019; Liu et al., 2021b; Hu et al., 2021) to allow the efficient management of weights for each task.

## 6   Conclusion

In this study, we introduce DRAFT, a simple but effective approach that first applies a dense retriever model for few-shot classification across diverse topics. Despite possessing 177 times fewer parameters than LLMs, DRAFT demonstrates superior performance in few-shot topic classification. These results imply the effectiveness of DRAFT in classifying a diverse array of real-world topics. We anticipate that DRAFT holds the potential to be implemented in practical contexts and actively contribute to addressing diverse societal challenges, including those encompassing specific topics.

## 7   Limitations

In Section 4.3, the negative query is defined as a text with content similar to the positive class but with a different topic. Although DRAFT demonstrates the ability to improve performance upon receiving a negative query, the automatic generation or construction of negative queries is necessary for real-world applications since people cannot manually provide a negative query for every topic.

Also, some ambiguous topics need to be calibrated in automatically constructed 291 datasets using the FactsNet website. For example, the queries in 'Sports' topic primarily revolve around baseball and golf. After training DRAFT, it can effectively classify content related to baseball and golf. However, after manually examining the test dataset for 'Sports' topic, it becomes apparent that examples of other sports, such as basketball and tennis, are belonged to the positive class. Although basketball and tennis undoubtedly fall under the sports category, the queries are composed solely of content related to specific examples of sports, such as baseball and golf. The ambiguity of the test dataset poses a challenge for DRAFT, proving to be a difficult task even for humans to classify these contents with the same few-shot samples accurately. Therefore, conducting human manual reviews for each of the 291 subtopics within FactsNet to filter out ambiguous topics could enhance the reliability of experimental results.

Moreover, a limitation of DRAFT lies in the need for mathematically rigorous proof for the validity of its MQR. As a result, in future research, we plan to undertake endeavors to address the quality issue of FactsNet and establish rigorous mathematical proof for evaluating the effectiveness of MQR.

## 8   Ethics Statement

We explicitly mention the copyright of the FactsNet dataset, crawled from the website. We firmly state that we use the Factset dataset simply for evaluation due to the absence of a benchmark dataset containing diverse topics. Also, we employ five annotators while building two additional manually constructed datasets in Section 4.3. Our annotators are affiliated with our company and receive compensation through wages for their labeling work. Detailed instruction for the annotation task is in Appendix A. Also, we strongly discourage any misuse of DRAFT for illegal activities.

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

## A  Details of datasets

| Major category | Data information | | |
|---|---|---|---|
| | Topic-num | Sample-num | Avg. token length |
| Lifestyle | 46 | 220 | 39.4 |
| History | 56 | 166 | 45.3 |
| Nature | 54 | 169 | 43.5 |
| World | 113 | 203 | 46.0 |
| Science | 22 | 214 | 44.0 |
| Total | 291 | 193 | 44.1 |

Table 5: The detailed information includes the 'Topic-num', 'Sample-num', and 'Avg. token length', which represent the number of subtopics, the average number of samples, and the average token count, respectively.

In this study, we use a total of six datasets. FactsNet is used in Section 4.2. Religion and South Korea are used in Section 4.3. AGNews, DBpedia, and TREC, are used in Section 4.4.

**FactsNet**  We manually construct 291 test datasets for few-shot topic classification tasks across diverse topics by crawling `http://www.facts.net`. It has a three-level hierarchical structure consisting of a major category, a subcategory, and a subtopic. The major category comprises six topics ('Lifestyle,' 'History,' 'Nature,' 'World,' 'Science,' and 'General'). However, we exclude 'General' when constructing the dataset since it consists of ambiguous content that does not align well with topic classification. Detailed information for five major categories can be found in Table 5. Table 6 provides types of subcategories under each major category and all subtopics under each subcategory. In constructing the dataset, we set the positive class as 0, the easy negative class as 1, and the hard negative class as 2. The composition of the label is: (0:Positive, 1:Negative), where the positive class belongs to content related to the subtopic, and the negative class contains easy negative and hard negative. Table 7 shows example samples of datasets.

**Religion**  Unlike FactsNet dataset, we determine whether samples are appropriate for positive and negative classes by examining each sample individually. Five annotators label each sample to determine if it contains content related to the topics defined in Religion. Samples for the hard negative class are constructed with the same method. The instruction for annotators regarding all samples is as follows: *'Please choose the correct label for the following sentence. If the following sentence is related to Jewish or Islam, choose 0. If the fol-*

*lowing sentence is not related to religion, choose 1. If the following sentence is related to Buddha or Hinduism, choose 2. Answer: 0, 1, 2'*. The final label for all classes is determined through a majority vote. Religion dataset consists of 78 samples from `https://www.history.com/topics/religion`. The composition of the label is: (0:Positive, 1:Negative), where the negative class contains both easy negative and hard negative. Examples of the dataset can be seen in Table 8.

**South Korea**  For South Korea, the test dataset consists of 112 samples from `http://www.facts.net`, `https://pitchfork.com`, `https://www.britannica.com`, and `https://en.yna.co.kr`. This dataset is constructed using a similar method to Religion dataset. Positive samples are related to 'South Korea'. Hard negatives are composed of content related to other Northeast Asian countries, such as 'North Korea', 'China', and 'Japan', which fall under the same country category but differ from South Korea. The instruction for annotators regarding all samples is as follows: *'Please choose the correct label for the following sentence. If the following sentence is not related to the country, choose 1. If the following sentence is related to South Korea, choose 0. If the following sentence is related to North Korea, China, or Japan, choose 2. Answer: 0, 1, 2'*. The composition of the label is: (0:Positive, 1:Negative). Examples of the dataset can be seen in Table 9.

**AGNews**  A collection of news articles from ComeToMyHead is used for a 4-way topic classification problem. We use a test dataset consisting of 7,600 samples from `https://huggingface.co/datasets/ag_news`. The composition of the label is: (0:World, 1: Sports, 2:Buisness, 3:Sci/Tec).

**DBpedia**  DBpedia ontology classification dataset is used for a 14-way topic classification problem. We use a test dataset consisting of 70,000 samples from `https://huggingface.co/datasets/dbpedia_14`. The composition of the label is: (0:Company, 1:EducationalInstitution, 2:Artist, 3:Athlete, 4:OfficeHolder, 5:MeanOfTransportation, 6:Building, 7:NaturalPlace, 8:Village, 9:Animal, 10:Plant, 11:Album, 12:Film, 13:WrittenWork).

**TREC**  Text Retrieval Conference Question Classification is used for 6-way question classification tasks, which differ from topic-based content classification. The sample consists of a question, and

the labels are divided into six types of question answers. A test dataset consisting of 500 samples is used from `https://huggingface.co/datasets/trec`. The composition of the label is: (0:Abbreviation, 1:Entity, 2:Description and abstract concept, 3:Human being, 4:Location, 5:Numeric value).

# B  Details of experiment

## B.1  Resources

When training a dense retriever model in DRAFT, a bi-encoder, we employ a Distributed Data Parallel (DDP) setting with three GPUs. However, we only use a single GPU to train the classifier for the topic classification task in DRAFT.

During the experiments with LLMs, we use OpenAI's API [4] and conduct ICL. The cost for GPT-3 and InstructGPT models, with a model size of 175B, is $0.02 per thousand tokens, while the cost for a model size of 2.7B is $0.0004 per thousand tokens. Due to API costs, we run the LLMs experiments directly only in Section 4.2, and in Section 4.4, we take the results from (Zhao et al., 2021). Therefore, the candidates for k in the k-shot setting differ between them.

## B.2  Hyperparameters

Hyperparameter settings for training a bi-encoder are identical to those of (Karpukhin et al., 2020). In all experiments, all classifiers in DRAFT use a fixed batch size of 256 and a maximum token length of 128. As mentioned in Section 4.4, the subspace and number of queries are tuned using grid search for hyperparameter search, using accuracy as the criterion, while the remaining hyperparameters are not tuned. The values used for the subspace size and the number of queries are [1, 5, 10, 50, 100, 500, 1000] and [5, 10, 15, 20, 25, 50], respectively. The samples extracted from the training datasets of benchmark datasets in Section 4.4 are randomly selected using five random seeds: [1234, 5678, 1004, 7777, 9999].

## B.3  Prompts format

We refer to (Chiu et al., 2021) for the prompt format on LLMs for ICL in Section 4.2. Table 11 lists examples of prompt formats. The prompt format for LLMs in Section 4.4 can be found in (Zhao et al., 2021).

## B.4  Query examples

In Section 4.2, the queries obtained from DRAFT, sourced from Quick Facts, are additionally utilized as ICL examples for the LLMs. On the other hand, in Section 4.3, five annotators formulate both positive and negative queries relevant to defined classes in Religion and South Korea. Queries used in Section 4.2 and Section 4.3 can be seen in Table 10. For Section 4.4, we select queries randomly from the training dataset, using the random seed referred to in Appendix B.2.

---

[4] `https://openai.com/api/pricing/`

| Major category | Subcategory | Subtopic |
|---|---|---|
| **Lifestyle** | Entertainment | Books, BTS, Christmas Songs, Disney, Entertainment, Friends, Harry Potter, Lego, Lord Of The Rings, Marvel, Minecraft, My Cousin Vinny, Netflix, Pokemon, Rock Paper Scissors, Spotify, Star Wars, Tiktok, Video Game |
| | Food | Avocado, Beer, Breakfast Around The World, Chocolate, Coca-Cola, Coffee, Cornflakes, Egg, Food, Ice Cream, McDonald, Nutella, Pineapple, Pizza, Pizza Hut, Starbucks, Strawberry, Watermelon |
| | Health | Health |
| | Sports | Baseball, Basketball, ESPN, Golf, Running, Soccer, Sports |
| **History** | Culture | Egyptian Gods and Goddesses, Hispanic Culture, Language, St. Patrick's Day, Thanksgiving, The Mayans, Valentine's Day |
| | Historical Events | 4th Of July, 911, Boston Tea Party, Cinco de Mayo, Historical Events, Korean War, Pearl Harbor, World War 1, World War 2 |
| | People | Alexander the Great, Andre the Giant, Aphrodite, Auschwitz, Barack Obama, Beethoven, Bill Gates, Bruce Lee, Cleopatra, Confucius, Donald Trump, Florence Nightingale, George Washington, Hades, Hera, Jesus, Julius Caesar, Marie Curie, Marilyn Monroe, Martin Luther King Jr., Maya Angelou, Michael Jackson, Michael Jordan, Mother Teresa, Mozart, Neil Armstrong, Oprah Winfrey, Pablo Escobar, People, Poseidon, Pythagoras, Steve Jobs, The Beatles, US Presidents, William Shakespeare, Women Leaders |
| | Religion | Bible, Buddhism, Easter |
| **Nature** | Animals | Animal, Bat, Bear, Bee, Black Bear, Bobcat, Bug, Camel, Cat, Chicken, Chihuahua, Corgi, Cow, Dolphin, Elephant, Fish, Frog, Golden Retriever, Hippo, Horse, Killer Whale, Lion, Little Red Flying Fox, Lobster, Megalodon, Monarch Butterfly, Monkey, Otter, Owl, Parasite, Penguin, Pigeon, Rabbit, Red Tailed Hawk, Shark, Siberian Husky, Snake, Tiger, Starfish, Whale, Wolf |
| | Human Body | Baby, Human Body, Redhead, Sex |
| | Plants | Plant, Sunflower |
| | Universe | Earth, Jupiter, Moon, Northern Lights, Pluto, Saturn, Universe |
| **World** | Cities | Barcelona, Berlin, City, Galapagos Islands, London, New York, Paris, Pompeii, Tokyo, Venice |
| | Countries | Afghanistan, Ancient Egypt, Ancient Greece, Ancient Rome, Argentina, Belgium, Bolivia, Brazil, Canada, China, Colombia, Costa Rica, Country, Denmark, Dominican Republic, Ecuador, Egypt, Greece, Germany, Greenland, Guatemala, Haiti, Honduras, Hong Kong, Iceland, India, Iran, Israel, Italy, Japan, Mexico, Mongolia, New Zealand, Nigeria, North Korea, Norway, Pakistan, Philippines, Poland, Portugal, Puerto Rico, Russia, Scotland, Singapore, South Korea, Spain, Sweden, Switzerland, Thailand, Turkey, Venezuela, Vietnam |
| | Landmarks | Area 51, Big Ben, Buckingham Palace, Burj Khalifa, Colosseum, Eiffel Tower, Golden Gate Bridge, Grand Canyon National Park, Machu Picchu, Mount Rushmore, Niagara Falls, Notre Dame Cathedral, Panama Canal, Statue of Liberty, Stonehenge, Sydney Opera House, Victoria Falls, Winchester House, Yellowstone National Park, |
| | US States | Alaska, Connecticutm, Hawaii, Illionis, Indiana, Kansas, Kentucky, Las Vegas, Louisiana, Maine, Maryland, Michigan, Minnesota, Missouri, Montana, Nebraska, Nevada, New Hampshire, New Mexico, New York State, North Carolina, Ohio, Oregon, Pennsylvania, Rhode Island, South Carolina, Texas, US States, Virginia, Virginia Plan VS New Jersey Plan, Wisconsin |
| **Science** | Biology | Bacteria, Biology |
| | Chemistry | Chemistry, Gold, Lithium |
| | Geography | Deforestation, Desert, Geography, Great Pacific Garbage Patch, Rainforest, Taiga Biomes, Weather |
| | Physics | - |
| | Technology | Airbnb, Amazon, Apollo 11, Car, Internet, LinkedIn, Paypal, Technology, WhatsApp, Yahoo |

Table 6: The major categories, subcategories, and subtopics in the FactsNet dataset, which follows a three-level hierarchical structure, are as follows. There are 291 topics in the subtopic column, and each subtopic has its unique test dataset. The subtopics are listed in alphabetical order.

| Topic | Text | Class |
|---|---|---|
| **Poseidon** | Many sculptures portray Poseidon with curly hair and a beard. Since he is the god of the sea, Poseidon was also depicted with a wet look distinguished by his hair. | P |
| | Aside from entanglements with each other, the gods have also been known to pursue affairs with humans and animals alike. | P |
| | Contrary to popular belief, sharks do have ears, although they aren't visible like most species. | N(EN) |
| | These interesting fun facts include a shark's similarities to cats. | N(EN) |
| | The Danish politician took office in 2011. But even when Helle was still in high school, she was already taking part in antiapartheid efforts and peace movements. | N(HN) |
| | Other organizations she served as chair to are the Congressional Black Caucus, Urban Affairs Committees, and Veteran's Affairs and Banking. | N(HN) |
| **China** | Just as the Han dynasty entered a period of political turmoil, epidemics and viruses began to weaken the Han Dynasty. | P |
| | From 1279 to 1368, the Yuan dynasty's vast size resulted in a more widespread foreign trade. | P |
| | Since then, the website's services had grown from air beds and shared spaces to higher-end properties including the houses, apartments, private rooms, and other properties. | N(EN) |
| | Unlike Accoleo's acquisition where Airbnb bought the company before putting up an office in Germany, Airbnb established its 2nd international office in London before it acquired Crashpadder. | N(EN) |
| | Wheat made up the biggest part of Ancient Egypt's harvests, used to make bread of various kinds. | N(HN) |
| | Jewelry, in particular, provided a colorful contrast to the usual plain appearance of Ancient Egyptian clothing. | N(HN) |

Table 7: 'Poseidon' and 'China' are examples of the 291 topics used in the few-shot topic classification task.

| Class | Text |
|---|---|
| P | Their God communicates to believers through prophets and rewards good deeds while also punishing evil. |
| P | Muslims are monotheistic and worship one, all-knowing God, who in Arabic is known as Allah. |
| EN | Enter your destination & your Tesla will automatically include Supercharging stops in your route. |
| EN | Comments section of Yahoo controlled by alt-reality biased moderators supporting lies harmful to the Nation. |
| HN | The religion's founder, Buddha, is considered an extraordinary being, but not a god. |
| HN | Hinduism is unique in that it's not a single religion but a compilation of many traditions and philosophies. |

Table 8: Examples from the test dataset under Religion dataset. During evaluation, easy negative (EN) and hard negative (HN) in Class column are used as the same negative class. We only use positive (P) and negative (N) labels for evaluation.

| Class | Text |
|---|---|
| P | Bong Joon Ho's Parasite made history for bagging 3 awards at the 2020 Oscars, which was the most of any film nominated. |
| P | Hangul classifies as one of the Altaic languages, is affiliated to Japanese, and contains some Chinese loanwords. |
| EN | Recently after more than 20 years as a Google account holder my YouTube channel was suspended without warning and without any reason given. |
| EN | Jordi Cruyff has signed his contract as FC Barcelona's new sporting director of football. He has already been serving in the role since July 1. |
| HN | The greatest health threat in North Korea is hunger. |
| HN | The Huizhou Ancient Town is a famous historical and cultural city in southern Anhui Province with over 2000 years of history. |

Table 9: Examples of the manually constructed dataset related to South Korea.

| Dataset | Query | Class |
|---|---|---|
| **Poseidon** | Poseidon is the Greek god of the sea. | Positive |
| | He is also the Greek god of storms, horses, and earthquakes. | Positive |
| | Poseidon's most famous brothers are Hades and Zeus. | Positive |
| | His parents are known as Cronus and Rhea. | Positive |
| | Poseidon is one of the twelve Olympians included in ancient Greek religion and mythology. | Positive |
| **China** | China's population is around 1.4 billion. | Positive |
| | China hosted the 2008 Summer Olympic Games in Beijing. | Positive |
| | China's population is 4 times larger than the United States. | Positive |
| | 60.7% of the population in China lives in urban areas. | Positive |
| | The Chinese civilization dates back to 7,000 BC. | Positive |
| **Religion** | Judaism is the world's oldest monotheistic religion, dating back nearly 4,000 years. | Positive |
| | Christianity is the most widely practiced religion in the world, with more than 2 billion followers. | Positive |
| | Islam is the second largest religion in the world after Christianity, with about 1.8 billion Muslims worldwide. | Positive |
| | Buddhism is a faith that was founded by Siddhartha Gautama ("the Buddha") more than 2,500 years ago in India. | Negative |
| | Hinduism is the world's oldest religion, according to many scholars, with roots and customs dating back more than 4,000 years. | Negative |
| **South Korea** | 'South Korea', 'K-pop', 'Seoul', 'Kimchi' | Positive |
| | 'North Korea', 'Japan', 'China' | Negative |

Table 10: Examples of queries for manually constructed datasets. 'Poseidon' and 'China' are example subtopics taken from the 291 subtopics in FactsNet. They only use positive class since tasks in Section 4.2 are one-class classification tasks. On the other hand, in Section 4.3, Religion and South Korea accept negative queries as well, resulting in two types of labels. The queries in South Korea comprise individual words rather than complete sentences. It underscores DRAFT's independence from the text format type of query, showcasing its capability to accept input regardless of whether it is in a sentence or word format.

| | Input | Class |
|---|---|---|
| **ICL Format** | Is the following text related to *Poseidon*? Answer yes or no.

He is also the Greek god of storms, horses, and earthquakes. : yes
Poseidon's most famous brothers are Hades and Zeus. : yes
His parents are known as Cronus and Rhea.' : yes

Many sculptures portray Poseidon with curly hair and a beard. Since he is the god of the sea, Poseidon was also depicted with a wet look distinguished by his hair.: | Positive |

Table 11: Formats used in LLMs in Section 4.2. The example shows a 3-shot setting for ICL, which have a similar format in (Chiu et al., 2021). We first concatenate the problem description and examples, followed by the test sample. The 'Class' column represents the actual label of the test sample.