# OpenReview forum: "DRAFT: Dense Retrieval Augmented Few-shot Topic classifier Framework"
_EMNLP/2023/Conference — EMNLP 2023 Findings_

### Official Review · Reviewer_r9Me · 2023-08-04

**Typos Grammar Style And Presentation Improvements:** Please refer to the reasons to reject.
**Soundness:** 3

**Excitement:**

2: Mediocre: This paper makes marginal contributions (vs non-contemporaneous work), so I would rather not see it in the conference.

**Missing References:**

NA

**Paper Topic And Main Contributions:**

This paper aims to solve the task of arbitrary topic classification. The authors designed a simple framework, named DRAFT, to train a classifier for few-shot topic classification. DRAFT first uses a few examples for specific topics to construct a customized dataset by a MQR algorithm. Finally, they fine-tune a topics classifier based on the customized dataset. Experiments on widely used datasets and manually constructed datasets demonstrate the effectiveness of DRAFT.

**Questions For The Authors:**

Please refer to the reasons to reject.

**Reasons To Accept:**

1. This paper proposed a simple method to solve the topic classification task, which is very significant and suffers from the problem of limited data in real-world applications.
2. This paper conducted extensive experiments on benchmarks and manually constructed datasets to demonstrate the effectiveness of DRAFT.
3. The dataset's details and experiment settings are declared clearly.

**Reasons To Reject:**

1. Although I can understand the meaning of each paragraph in the introduction chapter, the cohesion between paragraphs can be strengthened, thus improving the fluency of the paper. Some expressions in this paper are confusing, such as positive samples, negative samples, and negative quires.
2. The novelty of this paper is limited. The retrieval augmented methods have been widely used in many NLP tasks, such as machine reading comprehension, dialog generation, questioning answering, text classification, and sentiment analysis. However, there are no related works about retrieval-augmented methods in this paper.
3. Although the task is meaningful in practice, the proposed method is common in other NLP tasks. Even some studies proposed new retrieval methods, such as joint learning-based retrieval and retriever pre-training for specific downstream tasks. What do you think is the biggest difference between your retrieval method and theirs? Or what is the improvement of the retrieval model you made to solve the topic classification?
4. The case of some phrases in the article is confusing. Why should the first letter “C” of Customized dataset be capitalized? Why is the first letters “c” and “f” of the Topic classifier framework not capitalized in the title? Similar problems need to be reconfirmed.
In Table 1, the improvement of DRAFT is limited, did the authors do a significance test? For example, DRAFT achieves 80.2 and InstructGPT 175B achieves 79.9.

**Reproducibility:**

3: Could reproduce the results with some difficulty. The settings of parameters are underspecified or subjectively determined; the training/evaluation data are not widely available.

**Reviewer Confidence:**

4: Quite sure. I tried to check the important points carefully. It's unlikely, though conceivable, that I missed something that should affect my ratings.

---

> ### Author Rebuttal · Authors · 2023-08-28
>
> We are grateful for your careful reading of our paper, and your feedback. Your insightful feedback has guided us in expanding and improving our paper. We prepared a detailed response to your questions.
>
> $\textbf{Q1}$ Although I can understand the meaning of each paragraph in the introduction chapter, the cohesion between paragraphs can be strengthened, thus improving the fluency of the paper. Some expressions in this paper are confusing, such as positive samples, negative samples, and negative quires.
>
> $\textbf{Reply}$ Thank you for your feedback regarding the cohesion between the paragraphs in the introduction. We recognize the importance of fluidity and will take steps to enhance the transitions and connectivity between paragraphs. Also, to mitigate any confusion regarding terminology within the manuscript, we provide clearer definitions of the terms.
>
> - Positive samples: These are datasets related to the target topic for classification, corresponding to the positive class in the Customized dataset.
>
> - Negative samples: These are datasets that do not relate to the target topic, representing the negative class in the Customized dataset.
>
> - Positive queries: These are the multiple queries used to extract positive samples related to the target topic. Through the MQR algorithm, these multiple queries are processed simultaneously.
>
> - Negative queries: In the default setting of DRAFT, negative samples in the Customized dataset are randomly sampled from Data Collection. However, as mentioned in Section 3.3 (Expanding the Capabilities of DRAFT), by providing queries for the hard negative class similar to positive queries as input to DRAFT, performance can be enhanced. These queries are referred to as negative queries.
>
> Furthermore, we will expand upon and further clarify any terms in the paper that may be potentially confusing to readers.
>
> $\textbf{Q2}$ The novelty of this paper is limited. The retrieval augmented methods have been widely used in many NLP tasks, such as machine reading comprehension, dialog generation, questioning answering, text classification, and sentiment analysis. However, there are no related works about retrieval-augmented methods in this paper.
>
> $\textbf{Reply}$ There exist various retrieval-augmented methodologies in recent NLP research. These methods include solutions for tasks such as fact retrieval and open-domain question answering which provide answers to questions (e.g., FiD, RAG, REALM), techniques that apply during inference to reduce perplexity in Language Modeling (e.g., kNN-LMs), and strategies that operate akin to memory for specific knowledge or dialogues (e.g., RETROPROMPT, KIF). To the best of our knowledge, all existing retrieval-augmented methods universally handle only a single query as the input to the retriever and subsequently execute various downstream tasks. (As you pointed out, we will incorporate various retrieval-augmented methods being studied in NLP tasks within the related work section. We are grateful for your notification on the missing references.)
>
> However, in our research, we introduce the MQR algorithm, which is the first to accommodate multiple-query inputs for the retriever. To address the functionality of the MQR algorithm: it's not merely a naive extension that takes a single query input multiple times. Instead, it operates by considering the semantic similarities among the queries. We have validated the efficacy of the MQR algorithm through experiments, specifically by examining the performance of a classifier trained on a Customized dataset constructed using MQR. All experimental results presented in our paper (Table 1 in Section 4.2, Table 3 in Section 4.3, and Table 4 in Section 4.4) consistently demonstrate the significant performance of DRAFT, indirectly affirming the validity of MQR. Also, by referencing the additional experiment results in Table below, we can observe that the SimCSE, which was only finetuned on few-shot samples without undergoing the MQR process in DRAFT, exhibits notably low performance which highlights the efficacy of MQR.
>
>   | Major category | DRAFT | SimCSE | Random | Noun. | Dense. |
>   |----------|----------|----------|----------|----------|----------|
>   | Lifestyle  | 80.5(1)  | 67.1(3)  | 50.2(5) | 69.7(2)  | 58.0(4)  |
>   | History  | 84.7(1)  | 66.5(3)  | 49.4(4)  | 69.3(2)  | 42.5(5)  |
>   | Nature | 79.6(1)  | 67.0(3)  | 50.8(5)  | 62.9(4)  | 71.6(2)  |
>   | World | 82.1(1)  | 66.7(2)  | 50.2(4)  | 56.7(3)  | 45.3(5)  |
>   | Science  | 80.2(1)  | 66.7(2)  | 48.5(5)  | 54.5(4)  | 58.7(3)  |
>   | $\textbf{Avg. Rank}$ | $\textbf{1.0}$ | 2.6  | 4.6  | 3.0  | 3.8  |
>
> Upon examining Figure 2 in our manuscript, which presents an ablation study based on the number of queries in DRAFT, a clear positive correlation between DRAFT's performance and the number of queries is evident. This suggests that utilizing multiple queries as inputs proves more beneficial than relying solely on a single query. If DRAFT were to employ a naive retriever that can only handle a single query, as opposed to our MQR, the number of queries used to construct the Customized dataset would be limited to one, consequently adversely affecting its performance potential. While the MQR algorithm may seem straightforward at first glance, it becomes indispensable when dealing with multiple queries concurrently, which we regard as a distinctive novelty in our work. Furthermore, the MQR algorithm can be universally applied across all the retrieval-augmented studies previously mentioned. We consider the introduction of this straightforward idea – being able to process multiple queries simultaneously as input for retrieval – as one of the primary contributions of our research.
>
> $\textbf{Q3}$ Although the task is meaningful in practice, the proposed method is common in other NLP tasks. Even some studies proposed new retrieval methods, such as joint learning-based retrieval and retriever pre-training for specific downstream tasks. What do you think is the biggest difference between your retrieval method and theirs? Or what is the improvement of the retrieval model you made to solve the topic classification?
>
>
> $\textbf{Reply}$  Existing research on retrievers has predominantly focused on enhancing the performance of the retriever itself, often through approaches like joint learning or pretraining. However, all of these existing retrieval studies have remained confined to accepting only a single query as input and retrieving semantically relevant passages. In our study, we introduce the MQR algorithm, which is designed to take multiple queries simultaneously as input and leverages the similarities between these queries to extract higher quality passages. By expanding the number of queries that retrieval can process at once through MQR, we have enabled the performance of a new task: the one-class few-shot topic classification task, which was previously unattainable using traditional retrievers. Our ablation study results presented in Figure 2 of the manuscript demonstrate experimentally how classification performance improves as the number of queries increases.
>
> Additionally, the main experiment in Section 4.2 (Few-shot topic classification task) addresses a one-class few-shot topic classification task, wherein only one class related to the desired topic is defined. In the manuscript, we note that not only are there no existing retrieval-augmented few-shot classification methodologies suitable for this task, but there are also no standard few-shot classification models that can be applied to a one-class few-shot topic classification task. To the best of our knowledge, aside from the general-purpose model LLM, DRAFT is the pioneering approach capable of addressing this task. DRAFT introduces a straightforward methodology that employs a retrieval model for one-class few-shot topic classification tasks. Although it is a simple approach, there have been no prior attempts to employ retrieval for topic classification tasks. Our introduction of the MQR algorithm and its application represents a significant contribution of our research.
>
>
> $\textbf{Q4}$ The case of some phrases in the article is confusing. Why should the first letter “C” of Customized dataset be capitalized? Why is the first letters “c” and “f” of the Topic classifier framework not capitalized in the title? Similar problems need to be reconfirmed. In Table 1, the improvement of DRAFT is limited, did the authors do a significance test? For example, DRAFT achieves 80.2 and InstructGPT 175B achieves 79.9.
>
>
> $\textbf{Reply}$ Thank you for pointing out the crucial aspects of our paper. Our responses to each comment are provided below.
>
> - The term "Customized dataset" refers to a training dataset that is tailored for classifying specific topics, constructed by DRAFT accepting n queries as input. As it's a novel concept we introduced, we treated it akin to a proper noun, capitalizing the initial 'C'.
>
> - The reason for not capitalizing the 'c' and 'f' in the title was to represent the acronym DRAFT. However, in line with your feedback, we recognize the potential for confusion this might cause and will update them as capital letters.
>
> - Conducting a significance test on the 291 test data used in Table 1 was not feasible due to the high API costs associated with LLM. Nevertheless, in Section 4.2, both DRAFT and LLM utilize the same fixed query set in their experimental settings which assures no significant sources of randomness in the results. Moreover, the main point we aim to convey from the results in Table 1 is not that DRAFT significantly outperforms InstructGPT 175B, but rather that DRAFT achieves comparable performance with a smaller model. Therefore,we believe that there is no inconsistency in the overall logical flow of our argument in our manuscript.

---

### Official Review · Reviewer_dsxP · 2023-08-04

**Soundness:** 3

**Excitement:**

2: Mediocre: This paper makes marginal contributions (vs non-contemporaneous work), so I would rather not see it in the conference.

**Missing References:**

There are many existing few-shot text classification methods (even more for the classical text classification task).

Long-tailed Extreme Multi-label Text Classification by the Retrieval of Generated Pseudo Label Descriptions, Findings of ACL 2023

Contrastnet: A contrastive learning framework for few-shot text classification, AAAI 2022

Uncertainty-aware Self-training for Few-shot Text Classification, NeurIPS 2020

Hierarchical Attention Prototypical Networks for Few-Shot Text Classification, EMNLP 2019


Also, there is a retrieval augmented language model which may be directly comparable with the proposed model especially in an few-show learning with retrieval augmentation.

Atlas: Few-shot Learning with Retrieval Augmented Language Models, Arxiv 2022

**Paper Topic And Main Contributions:**

This paper addresses a topic classification task with arbitrary topics and limited labeled data. They use a dense retriever model to construct a customized dataset with a multi-query retrieval algorithm and fine-tune a pre-trained model for the classification task. Manually collected data sets and benchmark data sets are used for evaluation. The result shows that the proposed model is comparable with or outperforms the large language models with hundreds of billions of parameters. They argue that this is the first work to exploit a dense retriever model for a topic classification task.

**Questions For The Authors:**

Respectively for reasons to reject,

1.1. What is the novel part of this method? It seems that MQR (multi-query retrieval) is proposed by the authors, but I am wondering why this additional step is necessary if we can construct a customized data set for each topic directly by the MIPS algorithm.

1.2. Is a single classifier built for each topic? If then, can we still see it as more efficient than the LLMs with in-context learning since we need repeated dataset construction and model tuning by the number of topics.


2.1. Text classification has been long studied in Natural Language Processing and Data Mining community. It is a shame that there are not any discussion or comparison with these approaches. How is the proposed model different from the classical text classification methods?

2.2. Similarly, few-shot classification has been actively studied recently. While the authors compare ICL LLMs with the proposed models, they are more like general-purposed models not necessarily optimized for topic classification tasks. How is the existing few-show classification method different from this method?

2.3. Is it a fair comparison if only the proposed method exploits the external knowledge base (for data collection)? How much does it affect the performance improvement?


3.1. Besides, is there a particular reason why the authors constructed a manual data set for evaluation?



**Reasons To Accept:**

1. Easy-to-understand presentation and writing
2. Extensive evaluation and discussion
3. Higher performances than in-context large language models.


**Reasons To Reject:**

1. A straightforward implementation with a combination of existing works
2. Lack of comparison with existing (few-shot) text classification methods

**Reproducibility:**

5: Could easily reproduce the results.

**Reviewer Confidence:**

3: Pretty sure, but there's a chance I missed something. Although I have a good feel for this area in general, I did not carefully check the paper's details, e.g., the math, experimental design, or novelty.

---

> ### Author Rebuttal · Authors · 2023-08-28
>
> We're grateful for your detailed evaluation and insight on our research paper. We will make sure to incorporate the parts that you suggested for clarity and reflect their feedback on paper. Also, in addition to the papers you mentioned under missing references, we will incorporate other studies related to few-shot text classification in the related work section. We appreciate your pointing out the relevant research that we may have overlooked.
>
> $\textbf{Q1.1}$ What is the novel part of this method? It seems that MQR (multi-query retrieval) is proposed by the authors, but I am wondering why this additional step is necessary if we can construct a customized data set for each topic directly by the MIPS algorithm.
>
> $\textbf{Reply}$ As you pointed out, our proposed MQR algorithm, capable of processing multiple queries, operates differently from existing retrievers that solely utilize the MIPS algorithm for retrieving. To address the functionality of the MQR algorithm: it's not merely a naive extension that takes a single query input multiple times. Instead, it operates by considering the semantic similarities among the queries. Unlike scenarios such as open-domain question answering or fact retrieval where there's a definitive answer, when constructing a training dataset for topic classification, we posit that retrieving using multiple queries through MQR — rather than approximately retrieving relevant passages for a single query via the MIPS algorithm — leverages more information. This, in turn, enhances the quality of the retrieved passage for a specific topic. Experimentally, as illustrated in Figure 2 of Section 4.4, we've demonstrated a positive correlation between the number of queries and task performance, thereby validating the superiority of employing multiple queries over a single one, which directly uses MIPS algorithm, for the topic classification task using the MQR algorithm. If DRAFT were to employ a naive retriever that can only handle a single query, as opposed to our MQR, the number of queries used to construct the Customized dataset would be limited to one, consequently adversely affecting its classification performance.
>
> Although the MQR algorithm might appear straightforward, its importance is accentuated when processing multiple queries concurrently to ensure optimal performance. We consider this feature a distinctive novelty in our research. Furthermore, the MQR algorithm can be universally applied across all the retrieval-augmented studies previously mentioned. We consider the introduction of this simple and effective idea – being able to process multiple queries simultaneously as input for retrieval – as one of the primary contributions of our research. We will place greater emphasis on the stated contribution in the manuscript.
>
> $\textbf{Q1.2}$ Is a single classifier built for each topic? If then, can we still see it as more efficient than the LLMs with in-context learning since we need repeated dataset construction and model tuning by the number of topics.
>
> $\textbf{Reply}$   As you pointed out, DRAFT creates an independent classifier for each unique topic. When considering time efficiency, DRAFT takes longer to construct its classifiers since it needs finetuning stage compared to an API-utilizing LLM. However, in environments where API usage is restricted and only the model is compared, handling models of 175B size or more demands significant resources and costs, often burdensome for individual researchers or even for most companies. In contrast, DRAFT only requires three V100 GPUs in a local environment for building Customized dataset and training, while only one V100 GPU is sufficient for inference. Even in situations where ample GPU resources are available to run models like InstructGPT, additional processing and validation are required due to inherent biases in in-context learning, such as the majority label bias. Another reason we emphasize DRAFT's efficiency is its cost-effective approach. Unlike typical classifiers that need manually constructed training data in a supervised setting, DRAFT autonomously creates its training dataset (Customized dataset), presenting a more cost-efficient alternative.
>
> $\textbf{Q2.1}$ Text classification has been long studied in Natural Language Processing and Data Mining community. It is a shame that there are not any discussion or comparison with these approaches. How is the proposed model different from the classical text classification methods?
>
> $\textbf{Reply}$ We agree on the importance of addressing text classification methodologies in our paper. We will make sure to highlight the distinctions between DRAFT and other text classification methods in the related work section. General text classification methods predominantly utilize supervised learning, training classifiers to discern patterns in texts corresponding to each class. Similarly, DRAFT employs supervised learning from training data to construct its classifier. However, a primary distinction between traditional approaches and DRAFT lies in the construction process of training data. While most traditional studies rely on manually curated training data, incurring high human annotation costs, DRAFT efficiently and autonomously constructs its own training data using only a few samples corresponding to each class. In summary, while the objective functions during training are similar, DRAFT offers a more efficient and practical methodology for constructing the prerequisite training data, making it more applicable in real-world scenarios.
>
>
> $\textbf{Q2.2}$ Similarly, few-shot classification has been actively studied recently. While the authors compare ICL LLMs with the proposed models, they are more like general-purposed models not necessarily optimized for topic classification tasks. How is the existing few-show classification method different from this method?
>
> $\textbf{Reply}$ In our study, we emphasized the significance of accurately classifying long-tailed arbitrary topics in real-world scenarios. The primary experiment of our research is the task of classifying 291 topics as detailed in Section 4.2 (Few-shot topic classification task). This task differs from general topic classifications, which define two or more classes. Task in Section 4.2 is a one-class classification task where only a single topic is defined and tested. To the best of our knowledge, existing few-shot classification methods operate exclusively on tasks with two or more defined classes and cannot conduct in one-class classification tasks. In contrast, DRAFT, through our proposed MQR algorithm, can automatically construct a Customized dataset for training. This allows DRAFT to perform the classification task even when samples for only one topic are provided, showcasing its novelty over other few-shot classification methods.
>
> For quantitative assessment of our proposed framework, DRAFT, the best choice for a baseline model is the Large Language Model (LLM), and here's why: As you pointed out, while LLM is a general-purpose model and might not be the best for specific domain tasks, it is currently the only model capable of performing classification tasks across a variety of topics where only one class is defined (one-class few-shot topic classification). We will include a detailed explanation in our manuscript as to why the LLM is the most appropriate baseline for our main experiment: the one-class few-shot topic classification task.
>
> $\textbf{Q2.3}$ Is it a fair comparison if only the proposed method exploits the external knowledge base (for data collection)? How much does it affect the performance improvement?
>
> $\textbf{Reply}$ Using Data Collection is inherently a part of the framework we propose. Therefore, we believe it's challenging to discuss fairness when referencing external knowledge that doesn't directly contain the answers. To be precise, even the LLM model referred to more data during its pretraining stage and possesses a larger parameter size, giving it advantages compared to DRAFT. Given the absence of an entirely fair baseline model for the one-class few-shot topic classification task, we opted to use LLM in our experiments, as it represents the most suitable baseline for this particular task.
>
> As mentioned in the response to Question 2.2, there isn't an appropriate baseline to compare with DRAFT in a completely fair setting. Hence, our decision to use LLM, which demonstrates the strongest performance in NLP, as the baseline was the most optimal choice. To confirm the efficacy of the external knowledge base, we conducted additional experiments under the same conditions as Section 4.2, evaluating 291 datasets. In these experiments, we compared the performance of DRAFT when utilizing the external knowledge base to when it was trained solely on queries, excluding the external knowledge base. The Table below presents the results of an additional experiment where we finetuned SimCSE (same encoder type used in a classifier of DRAFT) using only a few-shot samples, without utilizing an external knowledge base (Data Collection). Setting aside the limited size of the training data, this task provided information only for the positive class corresponding to the target topic (a one-class sample). As a result, the classifier couldn't distinguish the negative class, leading to an extremely high recall and markedly low precision score. These results underscore the critical role of the external knowledge base, which enables the functioning of the MQR algorithm.
>
>   | Major category | DRAFT | SimCSE | Random | Noun. | Dense. |
>   |----------|----------|----------|----------|----------|----------|
>   | Lifestyle  | 80.5(1)  | 67.1(3)  | 50.2(5) | 69.7(2)  | 58.0(4)  |
>   | History  | 84.7(1)  | 66.5(3)  | 49.4(4)  | 69.3(2)  | 42.5(5)  |
>   | Nature | 79.6(1)  | 67.0(3)  | 50.8(5)  | 62.9(4)  | 71.6(2)  |
>   | World | 82.1(1)  | 66.7(2)  | 50.2(4)  | 56.7(3)  | 45.3(5)  |
>   | Science  | 80.2(1)  | 66.7(2)  | 48.5(5)  | 54.5(4)  | 58.7(3)  |
>   | $\textbf{Avg. Rank}$ | $\textbf{1.0}$ | 2.6  | 4.6  | 3.0  | 3.8  |
>
> $\textbf{Q3.1}$ Besides, is there a particular reason why the authors constructed a manual data set for evaluation?
>
> $\textbf{Reply}$  Recent studies in topic classification have primarily focused on improving algorithms or methodologies on benchmark datasets with a limited number of topics. However, when considering real-world applications, there's a demand for classifying topics that span hundreds of categories. While most benchmark datasets for topic classification are designed for tasks involving the classification among topics with two or more defined classes, the real-world scenario often deviates from this norm. For instance, in platforms overflowing with text information like social media, there's a significant demand to classify texts related to a specific topic (one-class topic classification). We believe that existing benchmarks do not adequately capture real-world applications. Therefore, in response to the aforementioned considerations, we took the initiative to construct an evaluation dataset comprising 291 topics, tailored for one-class topic classification, to more accurately reflect these real-world demands.

---

### Official Review · Reviewer_1bbv · 2023-08-11

**Typos Grammar Style And Presentation Improvements:** Why is the first letter of classifier…
**Soundness:** 3

**Excitement:**

3: Ambivalent: It has merits (e.g., it reports state-of-the-art results, the idea is nice), but there are key weaknesses (e.g., it describes incremental work), and it can significantly benefit from another round of revision. However, I won't object to accepting it if my co-reviewers champion it.

**Missing References:**

Similar methods applied for other NLP tasks:
[1] Generalization through memorization: Nearest neighbor language models.
[2] Decoupling knowledge from memorization: Retrieval-augmented prompt learning.
[3] Knowprompt: Knowledge-aware prompt-tuning with synergistic optimization for relation extraction.
[4] Recent advances in retrieval-augmented text generation.
[5] Augmenting transformers with knn-based composite memory for dialogue.
[6] Dense passage retrieval for open-domain question answering.
...

The references are suggested to be added in Related Work. They are some examples of retrieval-augmented methods in other NLP tasks.

**Paper Topic And Main Contributions:**

This paper proposes a framework designed to train a classifier for few-shot topic classification called DRAFT, which first applies a dense retriever model for few-shot classification across diverse topics. This approach demonstrates superior performance in few-shot topic classification than LLMs.

**Questions For The Authors:**

Refer to Reasons To Reject.

**Reasons To Accept:**

1. Meaningful tasks and innovative work. This work first applies a simple dense retriever model for few-shot classification across diverse topics. This work holds significant innovative value and achieves notably superior performance compared to LLM. The topic classification task in this paper has certain application values.
2. Detailed content. The introduction to the background, model design, experimental design, and other aspects are all written in a detailed and clear manner.
3. Well-structured experiments, with thorough consideration of experimental conditions. Datasets are diverse and comprehensive, and the experimental results and analysis are convincing.


**Reasons To Reject:**

1. The article only chooses LLM as the baseline for comparison. Despite having a large number of parameters, LLM may not necessarily achieve the best performance for tasks in specific domains. You are suggested to compare more baselines based on PTMs.
2. Despite effectiveness, the proposed model is a little simple, which has been popular in retrieval-augmented methods in NLP.
3. A case study or visualization study might give clearer explanations about why a dense retriever could improve the effectiveness.

**Reproducibility:**

4: Could mostly reproduce the results, but there may be some variation because of sample variance or minor variations in their interpretation of the protocol or method.

**Reviewer Confidence:**

4: Quite sure. I tried to check the important points carefully. It's unlikely, though conceivable, that I missed something that should affect my ratings.

---

> ### Author Rebuttal · Authors · 2023-08-28
>
> Thanks for your dedicated review of our work. Your critical feedback have helped us to extend and refine the paper. We provide a detailed response to your comments.
>
> $\textbf{Q1}$  The article only chooses LLM as the baseline for comparison. Despite having a large number of parameters, LLM may not necessarily achieve the best performance for tasks in specific domains. You are suggested to compare more baselines based on PTMs.
>
> $\textbf{Reply}$  In our study, we emphasized the significance of accurately classifying long-tailed arbitrary topics in real-world scenarios. The primary experiment of our research is the task of classifying 291 topics as detailed in Section 4.2 (Few-shot topic classification task). This task differs from general topic classifications, which define two or more classes. Task in Section 4.2 is a one-class classification task where only a single topic is defined and tested. To the best of our knowledge, existing few-shot classification methods operate exclusively on tasks with two or more defined classes and cannot conduct in one-class classification tasks. In contrast, DRAFT, through our proposed MQR algorithm, can automatically construct a Customized dataset for training. This allows DRAFT to perform the classification task even when samples for only one topic are provided, showcasing its novelty over other few-shot classification methods.
>
> For quantitative assessment of our proposed framework, DRAFT, the best choice for a baseline model is the Large Language Model (LLM), and here's why: While LLM is a general-purpose model and might not be the best for specific domain tasks, it is currently the only model capable of performing classification tasks across a variety of topics where only one class is defined (one-class few-shot topic classification). We will include a detailed explanation in our manuscript as to why the LLM is the most appropriate baseline for our main experiment: the one-class few-shot topic classification task.
>
> In Section 4.2, the baseline models used for the one-class few-shot topic classification task are detailed in Table 1, which includes LLM (GPT-3, InstructGPT), and in Table 2, which lists three weak baseline methods. As an extended experiment for this task, we further evaluated the SimCSE (same encoder type used in a classifier of DRAFT) , which was finetuned using few-shot samples that define only a single class. As anticipated, the results in the attached below Table show a lower performance due to the training being conducted on a dataset defined by a limited number of one-class samples. However, this additional experiment results with an extra baseline model has allowed us to underscore the excellence demonstrated by DRAFT as well as emphasize the difficulty of the one-class few-shot topic classification task. As shown in the Table below, we will integrate these results into Table 2 in our manuscript alongside the existing weak baselines.
>
>   | Major category | DRAFT | SimCSE | Random | Noun. | Dense. |
>   |----------|----------|----------|----------|----------|----------|
>   | Lifestyle  | 80.5(1)  | 67.1(3)  | 50.2(5) | 69.7(2)  | 58.0(4)  |
>   | History  | 84.7(1)  | 66.5(3)  | 49.4(4)  | 69.3(2)  | 42.5(5)  |
>   | Nature | 79.6(1)  | 67.0(3)  | 50.8(5)  | 62.9(4)  | 71.6(2)  |
>   | World | 82.1(1)  | 66.7(2)  | 50.2(4)  | 56.7(3)  | 45.3(5)  |
>   | Science  | 80.2(1)  | 66.7(2)  | 48.5(5)  | 54.5(4)  | 58.7(3)  |
>   | $\textbf{Avg. Rank}$ | $\textbf{1.0}$ | 2.6  | 4.6  | 3.0  | 3.8  |
>
>
> $\textbf{Q2}$  Despite effectiveness, the proposed model is a little simple, which has been popular in retrieval-augmented methods in NLP.
>
> $\textbf{Reply}$ There exist various retrieval-augmented methodologies in recent NLP research. These methods include solutions for tasks such as fact retrieval, open-domain question answering, and others which provide answers to questions (e.g., FiD, RAG, REALM), techniques that apply during inference to reduce perplexity in Language Modeling (e.g., kNN-LMs), and strategies that operate akin to memory for specific knowledge or dialogues (e.g., RETROPROMPT, KIF). To the best of our knowledge, all existing retrieval-augmented methods universally handle only a single query as the input to the retriever and subsequently execute various downstream tasks. (We will incorporate not only the papers you recommended but also cover various retrieval-augmented methods being studied in NLP tasks within the related work section. We are grateful for your suggestions on the missing references.)
>
> However, in our research, we introduce the MQR algorithm, which is the first to accommodate multiple-query inputs for the retriever. To address the functionality of the MQR algorithm: it's not merely a naive extension that takes a single query input multiple times. Instead, it operates by considering the semantic similarities among the queries. We have validated the efficacy of the MQR algorithm through experiments, specifically by examining the performance of a classifier trained on a Customized dataset constructed using MQR. All experimental results presented in our paper (Table 1 in Section 4.2, Table 3 in Section 4.3, and Table 4 in Section 4.4) consistently demonstrate the significant performance of DRAFT, indirectly affirming the validity of MQR. Also, by referencing the additional experiment results in Table presented in the Q1 Reply, we can observe that the SimCSE, which was only finetuned on few-shot samples without undergoing the MQR process in DRAFT, exhibits notably low performance which highlights the efficacy of MQR.
>
> Upon examining Figure 2 in our manuscript, which presents an ablation study based on the number of queries in DRAFT, a positive correlation between DRAFT's performance and the number of queries is evident. This suggests that using multiple queries as inputs proves more beneficial than relying solely on a single query. If DRAFT were to employ a naive retriever that can only handle a single query, as opposed to our MQR, the number of queries used to construct the Customized dataset would be limited to one, consequently adversely affecting its classification performance. Although the MQR algorithm might appear straightforward, its importance is accentuated when processing multiple queries concurrently to ensure optimal performance. We consider this feature a distinctive novelty in our research. Furthermore, the MQR algorithm can be universally applied across all the retrieval-augmented studies previously mentioned. We consider the introduction of this simple and effective idea – being able to process multiple queries simultaneously as input for retrieval – as one of the primary contributions of our research. We will place greater emphasis on the stated contribution in the manuscript.
>
> $\textbf{Q3}$  A case study or visualization study might give clearer explanations about why a dense retriever could improve the effectiveness.
>
> $\textbf{Reply}$ Table 8 and Table 9 in the Appendix provide examples of the Religion dataset and South Korea dataset, respectively, which were meticulously curated and reviewed by our research team on a sample-by-sample basis. In the Religion dataset, texts related to Jewish and Islam are designated as positive samples for our target topic, while texts associated with other religions, such as Buddha or Hinduism, are considered as hard negative samples. For the South Korea dataset, texts related to South Korea serve as the target topic's positive samples, with texts pertaining to North Korea, China, and Japan classified as hard negative samples. Given the similar semantics shared between the positive and hard negative samples in each dataset, it is a challenging task to perform fine-grained classifications.
>
> In Table 3 in Section 4.3, the M3 method, which employs both the MQR algorithm and a random sampling technique to construct negative samples for the target topic within the Customized dataset, demonstrates the highest performance. The M3 method, in comparison to the M1 method which constructs negative samples only with random texts from Data Collection, achieves a higher F1 score. The M3 method even exhibits greater accuracy when handling hard negative samples. This underscores the enhanced effectiveness of employing a dense retriever when constructing the Customized dataset. For instance, in constructing the negative dataset for the South Korea topic, the M3 method utilizing a dense retriever correctly classifies texts related to 'Kim Jong-il, the leader of North Korea,' as the Negative class, indicating they don't pertain to South Korea. On the other hand, the M1 method, which doesn't use a dense retriever, mistakenly categorizes them under the Positive class associated with South Korea, likely due to the significant interrelation between the two Koreas. Thus, the dense retriever plays a pivotal role in facilitating the assembly of a training dataset apt for fine-grained topic classifications, proving its efficacy.

---

### Meta-Review · Area_Chair_tTwv · 2023-09-16

**Recommendation:** 1

**Metareview:**

This paper proposes a framework called DRAFT for few-shot topic classification using a dense retriever model. The model is shown to outperform large language models (LLMs) in experiments, but the novelty of the approach is questioned by the reviewers as it is similar to existing retrieval-augmented methods. The paper is well-written and provides detailed content and thorough experimental results, but the reviewers feel that more comparison with other methods is suggested. The dataset construction method is also questioned as it uses a multi-query retrieval algorithm, and more details are needed to make it more clear. The reviewers also suggest more discussion and comparison with existing methods is needed.

---

### Decision · Program_Chairs · 2023-10-07

**Decision:**

Accept-Findings

**Comment:**

This paper proposes a framework called DRAFT for few-shot topic classification using a dense retriever model. The model is shown to outperform large language models (LLMs) in experiments, but the novelty of the approach is questioned by the reviewers as it is similar to existing retrieval-augmented methods. The paper is well-written and provides detailed content and thorough experimental results, but the reviewers feel that more comparison with other methods is suggested. The dataset construction method is also questioned as it uses a multi-query retrieval algorithm, and more details are needed to make it more clear. The reviewers also suggest more discussion and comparison with existing methods is needed.